# Characterization of Biogenic PbS Quantum Dots

**DOI:** 10.3390/ijms241814149

**Published:** 2023-09-15

**Authors:** Yoshiko Okamura, Ryo Shimizu, Yoriko Tominaga, Sachiko Maki, Tsunehiro Aki, Yukihiko Matsumura, Yutaka Nakashimada

**Affiliations:** 1Graduate School of Integrated Sciences for Life, Hiroshima University, Hiroshima 739-8530, Japan; aki@hiroshima-u.ac.jp (T.A.); nyutaka@hiroshima-u.ac.jp (Y.N.); 2Graduate School of Advanced Science of Matter, Hiroshima University, Hiroshima 739-8530, Japanytominag@hiroshima-u.ac.jp (Y.T.); 3Core Research for Evolutional Science and Technology (CREST), Japan Science and Technology Agency (JST), Tokyo 102-0076, Japan; mat@hiroshima-u.ac.jp; 4Consolidated Research for Biogenic Nanomaterials, Hiroshima University, Hiroshima 739-8530, Japan; sachiko.maki.d1@tohoku.ac.jp; 5Graduate School of Advanced Science and Engineering, Hiroshima University, Hiroshima 739-8527, Japan; 6Graduate School of Science, Hiroshima University, Hiroshima 739-8526, Japan

**Keywords:** biomineralization, biogenic nanoparticles, PbS, sulfide semiconductor

## Abstract

Heavy metals in a polluted environment are toxic to life. However, some microorganisms can remove or immobilize heavy metals through biomineralization. These bacteria also form minerals with compositions similar to those of semiconductors. Here, this bioprocess was used to fabricate semiconductors with low energy consumption and cost. Bacteria that form lead sulfide (PbS) nanoparticles were screened, and the crystallinity and semiconductor properties of the resulting nanoparticles were characterized. Bacterial consortia that formed PbS nanoparticles were obtained. Extracellular particle size ranged from 3.9 to 5.5 nm, and lattice fringes were observed. The lattice fringes and electron diffraction spectra corresponded to crystalline PbS. The X-ray diffraction (XRD) patterns of bacterial PbS exhibited clear diffraction peaks. The experimental and theoretical data of the diffraction angles on each crystal plane of polycrystalline PbS were in good agreement. Synchrotron XRD measurements showed no crystalline impurity-derived peaks. Thus, bacterial biomineralization can form ultrafine crystalline PbS nanoparticles. Optical absorption and current–voltage measurements of PbS were obtained to characterize the semiconductor properties; the results showed semiconductor quantum dot behavior. Moreover, the current increased under light irradiation when PbS nanoparticles were used. These results suggest that biogenic PbS has band gaps and exhibits the general fundamental characteristics of a semiconductor.

## 1. Introduction

The removal and recovery of heavy metals using microorganisms have long been proposed. Hazardous heavy metals can be absorbed and solidified through microbial biomineralization [1]. Microorganisms produce biominerals through reactions at ambient temperatures and pressures; therefore, biomineralization has a limited environmental impact and low process costs. Furthermore, due to specific biological selectivity, the recovery and reuse of minor and heavy metals from wastewater and oceans can be expected even at dilute concentrations. Studies examining metal sulfide precipitation as a strategy for bioremediation have primarily focused on removing metals from metal-contaminated wastewater, emphasizing the efficiency of metal removal rather than the precipitation process or end products. Recently, growing interest has shifted the focus not only to the bioremediation of metal-rich wastewater, but also to the effective utilization of the end precipitates. Especially, bio-fabricated semiconducting materials have been focused on metal sulfides [2]. The resulting metal sulfides are particularly important in industry due to their optical and electrical properties and finite global resources [3]. Sulfide semiconductors, such as cadmium sulfide (CdS) and lead sulfide (PbS), synthesized by microbes have already been explored [4,5]. These reports utilized yeast species, such as *Candida glabrata*, *Schizosaccharomyces pombe*, and *Toluropsis* sp., which feature short metal-chelating peptides with the general structure (Glu-Cys)_n_-Gly that control nucleation and form nanocrystallites intracellularly [4,5]. Recently, Wei et al. employed the L-cycteine-desulfurizing bacterium *Lysinibacillus sphaericus* SH72 to demonstrate the one-step synthesis of PbS nanocrystallites [6]. L-cycteine was not only a source of S^2−^ produced by cycteine desulfhydrase, but also worked as a stabilizer to coat the PbS nanoparticles [6]. In the early stage, CdS was spontaneously formed by adding cadmium chloride and sodium sulfide to a culture of *Escherichia coli* in the stationary phase [7]. Because *E. coli* does not have sulfate-reducing activity, sodium sulfide was added to provide a source of S^2−^. Interestingly, electron microscopy has revealed the presence of intracellular CdS nanocrystals in bacteria, suggesting that CdS formed intracellularly after the cells incorporated Cd^2+^ and S^2−^ [7]. Moreover, the authors hypothesized intracellular thiol or glutathione contents were related to nanosized control (2–5 nm) [7]. This report indicated that spontaneously formed CdS could be maintained as a nanosized particle due to the bacterial organic matters. In further studies using sulfate reducers, HS^−^ or S^2−^ accumulated during cultivation, and metal ions were injected into the cultures to form metal sulfides extracellularly, resulting in the synthesis of zinc sulfide (ZnS) and copper sulfide (CuS) nanoparticles (~5 nm) [8,9]. An amine (-NH_2_) has been detected among the synthesized ZnS nanoparticles using X-ray photoelectron spectroscopy, suggesting that the presence of amino acids maintained the nano-dimensional average crystallite size and increased crystallite aggregation [8]. These studies have demonstrated that the synthesis of metal sulfides by bacteria is reproducible and controllable; however, utilizing microbes with heavy metal tolerance is crucial.

PbS is a group IV–VI compound semiconductor formed from lead in group 14 (group IV) and sulfur in group 16 (group VI). Its energy band gap is 0.4 eV, and it has photoelectric sensitivity in the infrared region at around 3500 nm [10,11]. PbS was developed in the 1940s, primarily in Germany, for use in infrared detectors. Subsequently, it was studied more broadly and used in precision instruments such as photoconductive cells and infrared spectrophotometers. Recently, chemically synthesized PbS nanoparticles have been used in studies of quantum dot (QD) solar cells, with conversion efficiencies of 9.9% [12] and 10.7% [13] reported in 2015.

On the other hand, among the studies of PbS particles synthesized by microorganisms, *Torulopsis* sp. formed intracellular PbS nanoparticles 2–5 nm in diameter. These particles exhibited an absorption peak at ~330 nm, corresponding to an energy band gap of ~3.75 eV [5]. In addition, it has been reported that *Rhodobacter sphaeroides,* a proteobacterium, formed extracellular PbS nanoparticles with an average particle size of 10.50 ± 0.15 nm, and the particle size could be controlled by adjusting the culture time [14]. The PbS nanoparticles synthesized by the microorganisms could potentially be used as semiconductors; however, few studies have evaluated the semiconductor applications of these nanoparticles.

In this study, we screened bacteria able to synthesize PbS nanoparticles and characterized the crystallinity and semiconductor properties of the resulting nanoparticles.

## 2. Results and Discussion

### 2.1. Bacterial Compositions of Enrichment Cultures PS-1 and PS-2

Two enrichment cultures showing tolerance to 1 mM lead acetate were obtained as good candidates and designated PS-1 and PS-2. The PbS-forming bacteria presented with a black color derived from minerals, which allowed these bacteria to be clearly distinguished from other bacteria. In the absence of lead acetate, the enrichment cultures PS-1 and PS-2 appeared red, indicating the presence of photosynthetic bacteria. When supplemented with lead acetate, the culture color darkened in a lead-acetate-concentration-dependent manner (Figure 1). Both enrichment cultures could recover all lead ions when grown with 1 mM lead acetate. The amount of minerals extracted from the harvested cells was consistent with the theoretical yield.

After the enrichment cultures PS-1 and PS-2 were obtained, we attempted to isolate the PbS-forming bacteria by streaking both cultures onto RCVBN agar plates supplemented with sodium sulfate and lead acetate at a final concentration of 1 mM. Although single colonies were repeatedly streaked onto an agar plate, the isolates could not be obtained as a single bacterial species in liquid culture. Moreover, the PbS formation ability and Pb^2+^ tolerance decreased with repeated colony formation. Therefore, the enrichment cultures PS-1 and PS-2 were used as microbial consortia for forming PbS nanoparticles.

We performed bacterial composition analyses according to 16S rDNA sequences. PS-1 primarily consisted of the genus *Geotoga* (29.0%), *Alkaliphilus transvaalensis* (17.4%), genus *Pseudomonas* (13.6%), genus *Marichromatium* (10.6%), family Desulfovibrionaceae (10.1%), *Acetobacteroides* (Blvii28) (7.2%), *Shewanella algae* (7.1%), and genus *Cohaesibacter* (1.2%). PS-2 primarily consisted of *A. transvaalensis* (20.3%), *Geotoga* (18.7%), *Acetobacteroides* (Blvii28; 12%), family Ruminococcaceae (11.7%), *Pseudomonas* (10.4%), *Marichromatium* (7.8%), *S. algae* (6.1%), *Desulfovibrio* (5.7%), and *Cohaesibacter* (0.4%). Figure 2 shows the bacterial composition based on the family and the genus. Unknown genus names and non-isolated species are shown by family or order names.

Based on the genus-level identification, both consortia primarily contained obligate anaerobic bacteria, such as *Geotoga* [15], *A. transvaalensis* [16], Blvii28 (current name: *Acetobacteroides*, Su et al., 2014) [17], and *Desufovibrio* [18], including family Ruminococcaceae. Other known anaerobic bacteria included *Marichromatium* [19] and *Shewanella* [20]. Our culturing conditions allowed a small headspace in each tube, so the resulting in cultures were micro-aerobic rather than anaerobic in liquid medium. Therefore, the bacteria may have been able to form colonies on the agar covering the soft agar (the agar overlay or top agar) but been unable to grow in nitrogen-purged liquid medium on a conventional clean bench. Because *Pseudomonas* and *Cohaesibacter* [21] are facultative anaerobic bacteria, their presence may be necessary to consume the dissolved oxygen, allowing anaerobic bacteria to grow.

Among these microorganisms, *Geotoga* [15,22], *Marichromatium* [23,24,25], Desulfovibrionaceae, and *Desulfovibrio* [26] have been reported to reduce sulfates or elemental sulfur. For example, *Desulfovibrio* is a sulfate-reducing bacterium: electrons are transferred from its respiratory chain, and sulfate is reduced into sulfide ions according to reaction 1, allowing sulfide ions to react with lead ions (reaction 2). Because PbS is extremely insoluble, with a solubility product constant (Ksp) of 3.4 × 10^−28^ [27], when Pb^2+^ is supplemented at 1 × 10^−3^ M, every S^2−^ ion is able to immediately react with a Pb^2+^ ion.
SO_4_^2−^ + 8e^−^ + 8H^+^ → S^2−^ + 4H_2_O (1)
Pb^2+^ + S^2−^ → PbS↓ (2)

Sulfur- or sulfate-reducing bacteria require strictly controlled oxygen deprivation conditions, such as a rigid anaerobic glove box equipped with inert gas flow, which is expensive and difficult to maintain and use. However, our stable microbial consortia are easy to maintain, inexpensive, and show faster growth than axenic cultures. To determine whether the PbS products formed by these mixed cultures yielded biogenic materials of sufficient quality, the characteristics of the PbS products were investigated.

### 2.2. Localization and Morphology of PbS Nanoparticles

We observed the intact bacterial cells in the PbS-forming culture, as well as the localization of the minerals using transmission electron microscopy (TEM; Figure 3). PS-1 is shown in Figure 3A–C, and PS-2 is shown in Figure 3D–F. In PS-1 and PS-2, rod-shaped bacteria of various lengths were observed. Bacteria approximately 2 μm in size were associated with extracellular minerals (Figure 3A,D). Mineral particles approximately 22 nm in size were observed intracellularly (Figure 3B,E). Energy-dispersive spectroscopy (EDS) analysis showed that both intercellular and extracellular mineral particles contained Pb and S, confirming PbS formation (Figure 3C,F). However, the morphologies and sizes differed between extracellular and intracellular nanoparticles. Figure 3C,F show the EDS spectra corresponding to the intracellular nanoparticles in Figure 3B,E, respectively. The carbon peak was due to the cell membrane. Oxygen and phosphorus peaks derived from PO_4_^3−^ are also present in Figure 3C. The negative charge of phosphate attracts metal ions, forming precipitates [28]. The intracellular PbS nanoparticles were spherical, distributed over the cell, and had a uniform size of around 22 nm. As previously reported, the size of the intracellular PbS nanoparticles might be stably controlled by association with the intracellular matrix [4,5].

By contrast, extracellular nanoparticles occurred at a specific site on the cell membrane, and extremely small granules were pushed outward and aggregated (Figure 3A,D). As mentioned above, sulfate-reducing bacteria generate S^2−^ through respiration (reaction 1). The respiratory chain was localized on the cell membrane. Thus, extracellular PbS nanoparticles would be attached at a specific site (respiratory chain exiting membrane microdomain). The sizes of the extracellular PbS nanoparticles (average: 5 nm) corresponded well to those of ZnS and CuS nanocrystallites synthesized extracellularly by bacteria in previous reports [8,9].

The extracted nanoparticles were observed at a high magnification; the aggregates consisted of particles a few nanometers in diameter. Figure 4 shows that 10 nm microcrystals grew in the presence of 1 mM lead acetate, but 3.9–5.5 nm QD-sized particles grew in the presence of 500 µM lead acetate. Lattice images were clearly observed, and the electron diffraction pattern showed regularly spaced spots for single crystals and concentric rings for polycrystals. These crystallographic data indicated that the PbS nanoparticles have crystallinity and that the crystal was growing.

Thus, we considered the PbS formation mechanism. First, sulfate reduction (S^2−^) occurred on the cell membrane (reaction 1), and subsequently, PbS formed through an ionic reaction with Pb^2+^ (reaction 2). Second, single molecules of PbS aggregated and became stacked in a regular pattern, following the crystal system. A spherical shape might have been observed because the biomolecules surrounding PbS restrained the crystal growth direction, causing the PbS nanoparticles to form a polyhedral structure and forming a spherical shape. However, the extraction and purification processes would have eliminated these biomolecules, allowing the nanoparticles to be gradually stacked together and aligned their crystallographic axes, leading to further crystal growth. Crystals showing cubic forms were also observed and the XRD pattern showed galena.

Furthermore, the sizes of the bacterial PbS might vary depending on the available lead concentration, suggesting that the particle size could be controlled by adjusting the concentration of lead acetate.

### 2.3. Crystallinity of Bacterial PbS Nanoparticles Evaluated Using XRD

XRD patterns are unique because they depend on the arrangements of atoms and molecules within a given material. XRD analysis can be used to determine the lattice constants associated with the components of a given material. The XRD pattern of a thin layer of bacterial PbS indicated polycrystals with faces corresponding to PbS (111), (200), (220), and (311) (Figure 5). RCVBN medium contains several mineral ions, such as Mg^2+^, Fe^2+^, Mn^2+^, MoO_4_^2−^, Zn^2+^, and Cu^2+^, as trace elements. Similar to biomolecules, the presence of these metal compounds could also introduce impurities in PbS nanoparticles. Therefore, the purity of the PbS nanoparticles was confirmed using synchrotron-source XRD. In Figure 5, the diffraction pattern from bacterial PbS nanoparticles (Figure 5, red-color chart) shows good agreement with theoretical peaks of PbS, suggesting that there are no impurities in the bacterial PbS nanoparticles. The diffraction pattern (2θ = 25°, 30°, 43°, 40°, 53°, 62°, and 68°) matched that of galena (cubic system).

To confirm this result, we prepared cell-free, synthetic PbS by mixing lead acetate and sodium sulfide, which displayed many impurities (Figure 5, gray-color chart), but they could not be identified. The relative purity of the biogenic PbS nanoparticles compared with chemically synthesized PbS particles may be due to the gradual release of S^2−^ from the enzymatic reaction, which allowed S^2−^ to react with metal ions on the order of the lowest Ksp. By contrast, during the chemical synthesis of PbS, 1 mM of S^2−^ was made available in a single dose, allowing it to react with various ions.

### 2.4. Optical Characteristics of Bacterial PbS Nanoparticles Evaluated with Optical Absorption Measurements

As the bacterial PbS nanoparticles were extremely small, similar to QDs, the bacterial PbS nanoparticles were examined for QD-like characteristics. The electronic state of a bulk semiconductor mainly has two bands: a valence band and a conduction band. The band gap between these two bands is specific to the material. However, the electronic state becomes discrete when the material size is reduced to the nanometer range. The effective band gap changes depending on the particle size, resulting in different physical properties for the bulk material. This phenomenon is known as the quantum size effect [29]. The energy band gap increases at quantum sizes, and the optical absorption shifts to shorter wavelengths. Using a tungsten lamp, the optical absorption spectrum was measured for the thin layer of bacterial PbS used for the XRD measurements. Comparing the absorption spectra of the optical source and the thin layers of bacterial PbS revealed a partial absorption edge for the PbS nanoparticles from PS-1 and PS-2. The energy band gap (Eg) of ordinary PbS is approximately 0.4 eV [10,11]. When converted to a wavelength, an absorption edge was observed at 3100 nm. However, partial absorption edges were observed near 1000 nm and 1200 nm for the PbS nanoparticles obtained from PS-1 and PS-2, respectively (Figure 6A,B). A graph of the relationship between α^2^ and the energy was generated, and the value of Eg was calculated. The Eg values for PS-1 and PS-2 were approximately 1.04 eV and 0.80 eV, respectively (Figure 6C,D). These values are greater than those for bulk PbS, suggesting that the quantum size effect is responsible.

The quantum size effect allows the energy levels of electrons to reach higher levels to satisfy the boundary conditions at both edges of the quantum well, such that a smaller electron-well width (particle size) results in higher energy levels and a larger band gap. A graph of the relationship between the QD diameter and the absorbed wavelength for PbS showed that the particle size range for wavelengths of 1100 nm (PS-1) and 1500 nm (PS-2) was 5–10 nm, which agrees with the values for the bacterial PbS nanoparticles observed using TEM. Thus, the PbS nanoparticles formed by bacteria have QD-like properties.

### 2.5. Electric Characteristics of Bacterial PbS Nanoparticles Evaluated Using Current–Voltage Measurements

When the bacterial PbS nanoparticles formed by PS-2 were subjected to voltages in the range of ±1 V, current measurements of 1.66–24.76 pA were observed under illumination, whereas current measurements of 0.81–17.27 pA were observed without illumination, indicating that the bacterial PbS nanoparticles were conductive. Current values were higher with illumination than without illumination. When voltages ±3 V were applied, the behavior of I-V curves was similar to that of the currents observed for a typical semiconductor substrate, indium phosphide (InP; Figure 7A,B). These results confirmed that photoelectric conversion occurred, in which light was absorbed and converted to current due to the quantum size effect. Although the bacterial PbS required ±3 V to achieve obvious photoelectric conversion, ±1 V was sufficient for the InP substrate. Moreover, when current was plotted against voltage, the shape of the plot was symmetric at 0 V for InP, whereas the line of symmetry in the plot for the bacterial PbS was not located at 0 V (Figure 7A,B). In addition, when using extracted minerals from the cells grown with 5 mM lead acetate, the minerals contained cubic-shaped PbS with 200 nm on a side, indicating secondary crystal growth. Therefore, photoelectric conversion did not occur (Appendix A). The results suggest that a higher concentration of lead acetate can enhance crystal growth but not maintain nano-sized particles, and the particle size is crucial for the semiconductor properties. Furthermore, we hypothesize that bacterial PbS contains non-conductive biomolecules, such as proteins or carbohydrates, which require a greater voltage than the commercialized product. However, these results indicate that bioprocessing may represent a potential low-cost method for use in the fabrication of semiconductors, which requires limited energy consumption.

These data show that biogenic PbS nanoparticles synthesized in mixed cultures display good crystallinity and the general fundamental characteristics of a semiconductor. The PbS nanoparticles obtained in this work were relatively easy to disperse in water; however, they may be prone to secondary crystal growth when dried. For device-development applications, further investigations remain necessary to optimize the synthesis conditions, identify size-controlling molecules, and determine nanoparticle stability over time.

## 3. Materials and Methods

### 3.1. Enrichment Culture

Marine samples, including gravel, sediments, algae, and sponges, together with seawater, were collected from 15 different sites in 2014 and maintained in RCVBN medium [30] at pH 7.6 to enrich the bacteria. The first enrichment cultures were then supplemented with lead acetate (final concentration of 1 mM), and PbS-forming bacteria were enriched (second enrichment culture). Two enrichment cultures were obtained as good candidates and designated PS-1 and PS-2 in 2017. PS-1 originated from sponges collected in the Kojima Islands, Imabari City, Ehime Prefecture, and PS-2 originated from gravels collected from the open sea of Hibiki in Kitakyushu City, Fukuoka Prefecture. Detailed information of the biosamples have been deposited at DDBJ under the accession numbers SAMD00603096 and SAMD00603097, respectively. The consortia are stably maintained and provide stable quantities of PbS nanoparticles beginning in 2017.

For cultivation, 1 mL of seed culture was transferred into 7 mL of fresh RCVBN medium supplemented with 1 mM lead acetate and 1 mM sodium sulfate and cultured in completely filled 8 mL screw-capped tubes at 24 °C under continuous illumination (38 µmol/m^2^/s).

### 3.2. Bacterial Analysis Based on 16S rDNA Sequence

The V3-V4 regions of the 16S rDNA were amplified using the 1st-341f_MIX (5′-ACACTCTTTCCCTACACGACGCTCTTCCGATCT-NNNNN-CCTACGGGNGGCWGCAG-3′) and 1st-805r_MIX (5′-GTGACTGGAGTTCAGACGTGTGCTCTTCCGATCT-NNNNN-GACTACHVGGGTATCTAATCC-3′) primers provided by Bioengineering Lab. Co., Ltd. (Kanagawa, Japan). The PCR mixture contained 1× KOD Plus-Neo Buffer, 2 mM dNTPs, 25 mM MgSO_4_, 0.05 U/µL KOD Neo Polymerase (TOYOBO Co., Ltd., Osaka, Japan), 0.25 μM forward and reverse primer, and 2 µL genomic DNA. Genomic DNA was extracted from PS-1 and PS-2 using a NucleoSpin Microbial DNA kit (Takara Bio, Inc. Kusatsu, Shiga, Japan). Thermal cycling conditions were as follows: pre-denaturation for 2 min at 94 °C for one cycle, followed by 25 cycles of denaturation at 95 °C for 30 s, annealing at 55 °C for 30 s, extension at 68 °C for 30 s, and final extension at 68 °C for 5 min (T100^TM^ Thermal Cycler, Bio-Rad, Hercules, CA, USA). PCR products were subjected to 1% agarose gel electrophoresis to confirm the lengths of the products. After confirmation, PCR products were sent to Bioengineering Lab. Co., Ltd. (Kanagawa, Japan). PCR products were purified using AMPure XP (Beckman Coulter, Inc., Tokyo, Japan) to purify the PCR products, and Bioengineering Lab Co., Ltd. (Kanagawa, Japan) prepared an amplicon library for sequencing. Sequencing was performed with MiSeq (Illumina, Inc., San Diego, CA, USA) using a 2 × 300 bp paired-end run.

For data analysis, read-quality filtering was first performed. Only sequences for which the beginning of the read matched perfectly with the primer sequence were extracted using a FASTQ barcode splitter (FASTX toolkit ver. 0.0.14) [31]. Then, the primer sequence was trimmed from the extracted sequences. Sequences with a quality value of less than 20 were then removed using sickle tools (FASTX toolkit ver. 0.0.14), and sequences that were less than 150 bases in length were discarded, along with their paired sequences. Sequences that passed quality filtering were merged using the Paired End Merge script (FLASH ver. 1.2.11) [32]. Merging conditions were as follows: fragment length of 420 bases after merging, fragment length of 280 bases in reads, and a minimum overlap length of 10 bases. Sequences that passed all filtering steps were checked for chimeric sequences using USEARCH (https://www.drive5.com/usearch/; accessed on 29 January 2018). Taxonomic information was assigned using the Greengene database and the QIIME pipeline (ver. 1.9.0) [33]. The clustering of operational taxonomic units (OTUs) was set at 97%. OTU creation and phylogenetic estimation were performed using QIIME’s workflow script with no references and all parameters set to default.

### 3.3. PbS Nanoparticle Formation and Extraction

PbS nanoparticles were formed in cultures of PS-1 and PS-2 supplemented with 1 mM lead acetate and 1 mM sodium sulfate in completely filled 8 mL screw-capped tubes and placed at 24 °C under continuous illumination for 1 week. After 1 week, the bacterial cells were centrifugated at 10,000× *g* for 5 min. The harvested cells were washed 3 times with TE buffer (10 mM Tris, 1 mM EDTA) and resuspended in 300 µL TE buffer, including lysozyme (200 µg/mL) and SDS (0.2%). After mixing, the cell suspension was incubated at 65 °C for 5 min to disrupt the cells. After centrifugation at 10,000× *g* for 5 min, the supernatant was removed, and the collected nanoparticles were washed with Milli-Q water.

Chemically synthesized particles were prepared as a cell-free control by mixing 1 mM lead acetate and 1 mM sodium sulfide in RCVBN medium. The precipitates were harvested and washed with Milli-Q water.

### 3.4. Transmission Electron Microscopy and Energy-Dispersive Spectroscopy

Whole cells and extracted nanoparticles were observed using TEM. A 1 mL sample of culture was used to harvest cells, and the collected cells were washed and diluted with Milli-Q water. The diluted cells or the extracted nanoparticles were mounted on 150-mesh copper grids coated with collodion (Nisshin EM Co., Ltd., Tokyo, Japan).

The localization of the nanoparticles within whole cells and particle morphology were observed using a TEM (JEOL JEM-2010, Tokyo, Japan), and the elemental compositions of their nanoparticles were determined with EDS (JEOL, JED-2300T).

### 3.5. X-ray Diffraction

Extracted or chemically synthesized particles in suspension were mounted on glass slides and dried at room temperature under a vacuum to obtain a thin layer. The crystalline structures of the specimens were determined with thin-film XRD using an RINT2100 X-ray diffractometer (Rigaku Corporation, Tokyo, Japan), with 2θ between 20° and 70°, a slit of 0.80 mm, a sampling width of 0.050°, and a scanning speed of 3.5 degrees/minute. The detailed crystal structure was resolved using synchrotron-source powder XRD. The data were collected using a Debye–Scherrer camera with an imaging plate detector at the BL44B2 beamline at SPring-8.

### 3.6. Optical Absorption Measurements

Optical absorption measurements were performed on the same specimens as the XRD measurements using an HEP 3965 tungsten lamp (Thorlabs, Inc., Newton, NJ, USA), a SpectraPro HRS-300 spectrometer (Princeton Instruments, Inc., Trenton, NJ, USA), and a PyLoN-IR detector (Princeton Instruments, Inc., Trenton, NJ, USA). The wavelength ranged from 800 to 1600 nm. Measurements were recorded at 200 nm intervals, and the individual data points were graphed with connecting lines.

### 3.7. Energy Band Gap (Eg) Calculation

The absorption coefficient, α, was determined using Equation (3). I_1_ and I_0_ were substituted with the normalized intensities. The measurement results centered on the wavelength of the absorption edge were normalized by setting the maximum intensity to 1. The results of the corresponding light-source measurements were also normalized.
α = −(1/x) log (I_1_/I_0_)(3)
where: α: absorption coefficient [arb. unit]; x: distance which light travels in a medium (thickness of sample) [cm]; I_1_: measured intensity of sample (normalized) [arb. unit]; I_0_: measured intensity of light source (normalized) [arb. unit];

The energy E converted from the wavelength was determined using Equation (4).
E = hc/λ = 1240/λ(4)
where: E: light energy [eV]; h: Planck’s constant, 6.62607 ×10^−34^ [Js]; c: speed of light in a vacuum, 2.99792458 × 10^8^ [m/s]; λ: light-source wavelength [nm].

The absorption coefficient, α, and the light energy were calculated using Equations (3) and (4), and the relationship between α^2^ and the light energy was plotted. Here, actual thickness of PbS cannot be measured; therefore, the calculated values of α^2^ were not indicated on the *y*-axis. Eg was determined graphically by taking the point where the tangent at the highest slope of the plot intersected with the background.

### 3.8. Current–Voltage Measurements

The electrical characteristics of bacterial PbS nanoparticles were evaluated by measuring the current–voltage characteristics. A suspension of extracted nanoparticles was mounted onto a semi-insulating InP substrate (8-mm^2^). The oxide film was removed from the substrate using a 10 min treatment with 10% hydrofluoric acid, followed by drying in a desiccator at room temperature under vacuum. After drying, indium electrodes were soldered at each corner of the PbS layer on the InP substrate and annealed for 20 s at 330 °C in a ceramic electric tube furnace (Asahi Rika Seisakusyo, Chiba, Japan). A thick layer of PbS was prepared using a 2 mm thick silicon sheet to make a 5 mm^2^ well. The suspension of extracted nanoparticles was mounted on this silicon on a glass slide to prevent delamination from the InP substrate. Then, the thick layer of PbS was released from the glass slide, and indium electrodes were formed and annealed. The sample was affixed to an insulating silicon substrate using double-sided tape during the measurements. Voltage was applied between ±1 V and ±3 V, and the resulting current was measured. Measurements were obtained with and without illumination.

## 4. Conclusions

In this work, we used microbial consortia to form PbS nanoparticles, in contrast to previous reports that used established monoculture strains. Despite containing obligate anaerobic bacteria, the consortia were easy to maintain in micro-aerobic conditions. The bacterial consortia demonstrated the ability to synthesize stable extracellular PbS nanocrystallites ranging in size from 3.9 to 5.5 nm. The PbS nanoparticles did not contain impurities, despite synthesis with mixed bacteria. Moreover, the PbS nanoparticles showed semiconductor QD-like behavior. The production of PbS nanoparticles by bacterial consortia has the potential to provide new avenues for the fabrication of green semiconductors.

## Figures and Tables

**Figure 1 ijms-24-14149-f001:**
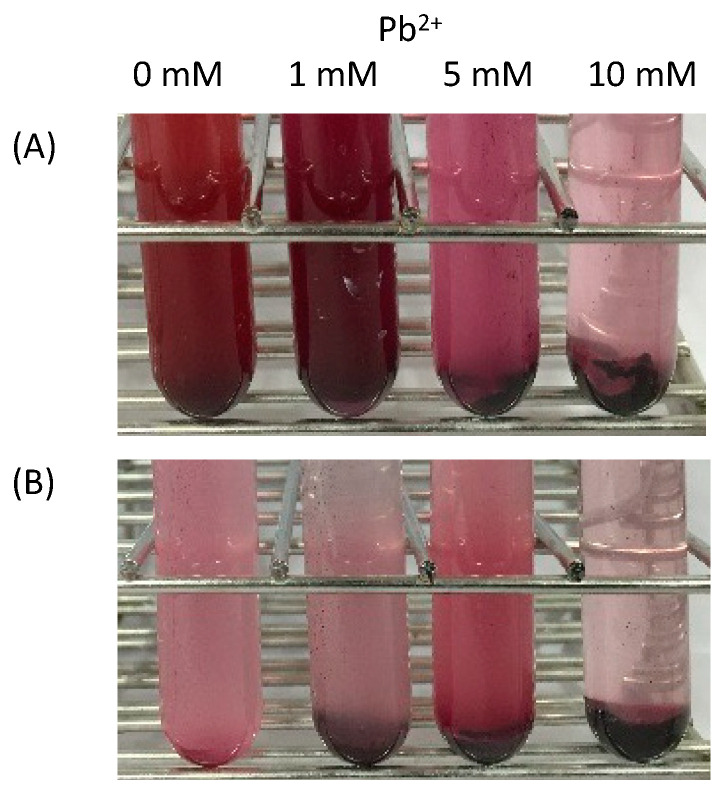
Changes in the culture color were dependent on the lead acetate concentration. (**A**) PS-1, (**B**) PS-2.

**Figure 2 ijms-24-14149-f002:**
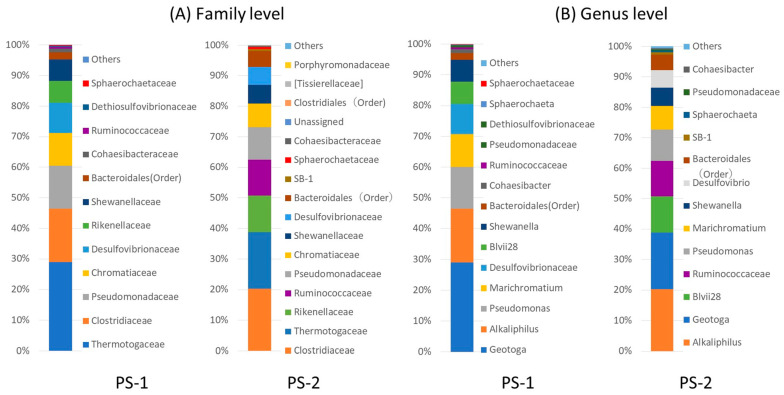
Relative abundance of bacterial taxa in the bacterial consortia PS-1 and PS-2 at the (**A**) family and (**B**) genus levels. Unidentified genera were replaced by the family or order name. The numbers of reads used for QIIME analysis were 277,786 and 32,756 for PS-1 and PS-2, respectively.

**Figure 3 ijms-24-14149-f003:**
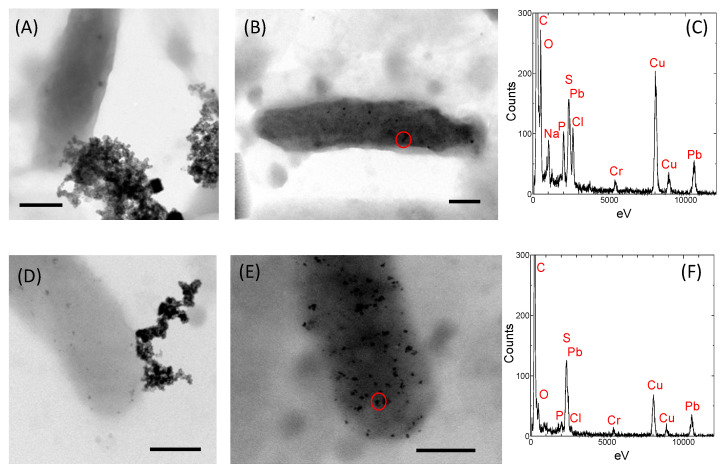
Transmission electron micrographs of PbS-forming bacteria (**A**,**B**,**D**,**E**) and the elemental compositions of the minerals (**C**,**F**). Scale bar: 500 nm. Circles in (**B**) and (**E**) indicate areas where the electron beam was applied for EDS.

**Figure 4 ijms-24-14149-f004:**
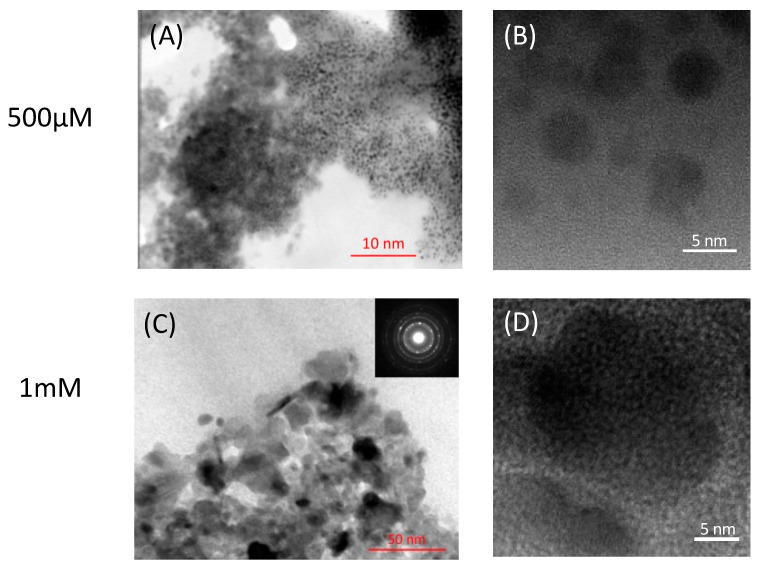
Transmission electron micrographs of the minerals extracted from PS-1 grown in the presence of (**A**,**B**) 500 µM or (**C**,**D**) 1 mM lead acetate. Insert in (**C**) indicates electron diffraction patterns of extracted minerals in (**C**).

**Figure 5 ijms-24-14149-f005:**
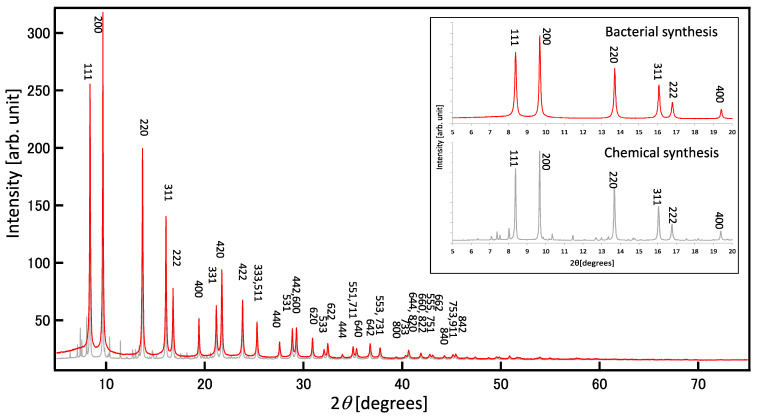
Synchrotron X-ray diffraction patterns. Red: the minerals extracted from PS-1 grown with 1 mM lead acetate. Gray: the precipitates prepared by mixing 1 mM lead acetate and 1 mM sodium sulfide. Insert: comparative patterns of 5–20 degrees.

**Figure 6 ijms-24-14149-f006:**
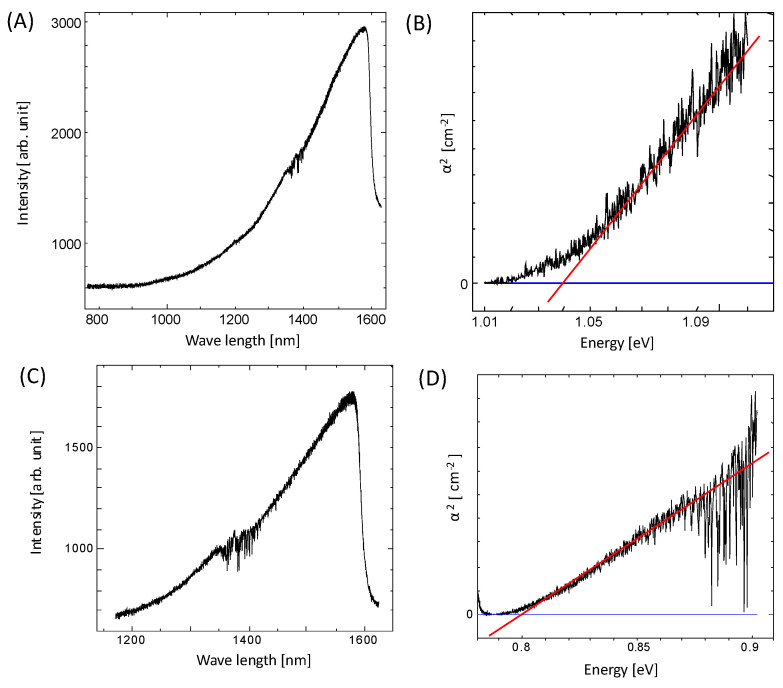
Measurement of (**A**,**C**) the optical absorption wavelength of bacterial PbS particles and (**B**,**D**) the relationship between α^2^ and the light energy. The intersection with the tangent line represents the energy band gap.

**Figure 7 ijms-24-14149-f007:**
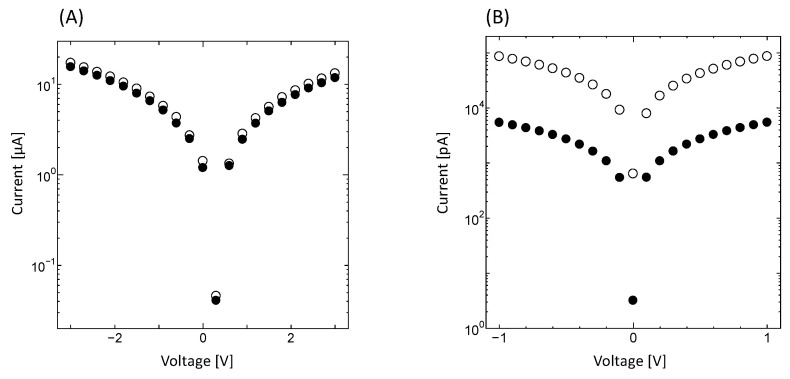
Current–voltage measurements of illuminated (**A**) bacterial PbS nanoparticles and (**B**) InP substrate. Open circle: with illumination, closed circle: without illumination. Voltages for PbS and InP were applied at ±3 V and ±1 V, respectively.

## Data Availability

Sequence data were deposited in DDBJ [accession number: PRJDB15813].

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
