# Peer review of "Characterization of Biogenic PbS Quantum Dots"

_ijms, 2023, doi:10.3390/ijms241814149_

Round 1

Reviewer 1 Report (New Reviewer)

This manuscript investigates the synthesis and characterization of biogenic lead sulfide nanoparticles. The approach seems to be innovative and the characterization is well-described. Following are my suggestions for further improvement:

1.       It would be helpful to describe PS-1 and PS-2 where they first appeared (Line 87).

2.       What is the yield for PbS nanoparticle production using this method?

3.       Section 2.2 showed that particle size can be controlled with lead concentration. It would be helpful to show the tunability of optical/electrical properties by varying lead concentration.

4.       How does the particle size distribution synthesized from PS-2 compare with PS-1?

5.       The use of “quantum dot”, “nanoparticle”, and “particle” needs to be more consistent. 

Author Response

To Reviewer 2

This manuscript investigates the synthesis and characterization of biogenic lead sulfide nanoparticles. The approach seems to be innovative and the characterization is well-described. Following are my suggestions for further improvement:

 Thank you for reviewing our revised manuscript again and for valuable discussion. We would like to respond point-by-point to the questions and the corrections are indicated in red letters in revised manuscript.

  1. It would be helpful to describe PS-1 and PS-2 where they first appeared (Line 87).

According to the suggestion, the sentences for explanation were added in L87-88.

  1. What is the yield for PbS nanoparticle production using this method?

Thank you for this comment. Pb2+ ions were not detected in the culture supernatant when grown at 1mM and the recovery rate of Pb2+ was 100% by the cells. Moreover, the amount of minerals extracted from the harvested cells was consistent with the theoretical yield. Therefore, we think all Pb2+ ions can be converted into PbS using PS-1 and PS-2. This explanation was added in L93-95.

  1. Section 2.2 showed that particle size can be controlled with lead concentration. It would be helpful to show the tunability of optical/electrical properties by varying lead concentration.

Thank you for your comments: we agree. Lower concentration tends to maintain the nanosized particles. In Fig. 4, secondary crystal growth after extraction from the cells was observed even at 1 mM. Larger cube crystals were observed in the cells grown at higher concentration. When using cube crystals with 200 nm on a side, µA level of currents were observed, however photoelectric conversion was not occurred. According to the advice, this observation was added in L282-287.

  1. How does the particle size distribution synthesized from PS-2 compare with PS-1?

We are sorry for the confusion we caused. Figure 3 showed both PS-1 and PS-2. This information was lacked. It was inserted in L. 155-156. As shown in Fig. 3, they formed similar sized nanoparticles. However, based on the results shown in Fig. 6, partial absorption edges were observed near 1100 nm and 1500 nm in PS-1 and PS-2, respectively, indicating the size is 5.7 nm and 8 nm.  

  1. The use of “quantum dot”, “nanoparticle”, and “particle” needs to be more consistent. 

Thank you for this comment. The PbS (particles) formed by PS-1 and PS-2 were indicated as “nanoparticles”. 

Reviewer 2 Report (New Reviewer)

My major problem with this study is described in lines 307 and 429. The Authors have used the “microbiota consortia” of an unknow origin. Therefore those experiments can’t be repeated by anyone else. Why the Authors haven’t used “standardized” consortia with well known composition and CFU. Further, if the Authors don’t compare their results with those obtained using single strains, how can we say whether this approach is effective or not?

Line 1, „Type of paper” should be removed

In the title, chemical formula “PbS” should be added

Reference [1] is over 20 years old, as it is the first reference that should introduce the reader with the topic. This reference should be replaced by a recent review paper (or a few).

References 3 and 4 are very old, i.e. ref. 4 is from 1989! A lot has changed since that time, and this should be updated.

Line 52, CdS would precipitate from the solution containing the Cd(II) and S(-II) ions anyway, so what is the benefit of adding the bacteria to that solution? It must be stated clearly in the introduction.

Line 77, it should be 10.50±0.15

The Ksp of lead sulfide is 8.0 × 10−28, so what’s the reason of adding the bacteria as this salt will precipitate anyway? Is it about the size of the crystals formed? This must be stated clearly in the introduction.

Line 106, it should not be Pb but Pb(II) or Pb2+

Line 151, what do you mean by “sufficient quality”?

Figure 3, C and F, please explain why the signals from Cu are more intensive than those from Pb? Where this Cu comes from?

Line 186, “Crystal lattice” can’t be observed, this is too colloquial. Please rephrase.

Why the pH was not monitored in those studies? This is crucial as it affects the formation of PbS.

Line 226, it should be “low purity”

Figure 5, why the Authors compare the regions between 2-theta 54 and 66? This region is not informative at all, due to the very low intensities (compared to the noise level). Instead, the Authors should compare the other range, 5-20 deg. Please provide “separate” grey and red patterns as I can see a lot of peaks at the grey (synthetic) one that do not appear at the red one. This must be explained as one may doubt that the correct phase of PbS has been obtained.

Line 449, please provide accession number

Author Response

Thank you for reviewing our revised manuscript and for valuable discussion. We would like to respond point-by-point to the questions and the corrections are indicated in red letters in revised manuscript. Correction parts were indicated in red letters.

My major problem with this study is described in lines 307 and 429. The Authors have used the “microbiota consortia” of an unknow origin.

Thank you for this comment. We have deposited detailed sample information in DDBJ and received accession numbers. According to the comments, the accession numbers for biosample were added in L324. 

Therefore those experiments can’t be repeated by anyone else. Why the Authors haven’t used “standardized” consortia with well known composition and CFU. Further, if the Authors don’t compare their results with those obtained using single strains, how can we say whether this approach is effective or not?

Thank you for your comments: we agree. However, as you also know, established strains and isolated species are less than 1% of total bacteria. Almost of them are difficult-to-cultivation. We have a lot of experiences that mix culture showed high performance but the isolates, even reconstruction of mix culture, did not show previous ability. Moreover, naturally occurred mix culture is easy to maintain the anaerobic bacteria as shown in Fig. 2 (more than 50%). Sources of PS-1 and PS-2 were obtained in 2014 and both enrichments were defined in 2017. Since then, all data in this study were obtained. And we also have obtained stable quantities of PbS nanoparticles for 6 years. The consortia have been maintained stably in RCVBN supplemented with 1mM lead acetate and transferred into fresh medium once a week. Please note that after 6 years, 2 consortia are still being maintained, and the major species have not changed. We would like to show the easy method to obtain mix cultures if the readers have difficulties in bacterial isolation. We have added the explanation of stability of consortia in L323-325.

Line 1, „Type of paper” should be removed

Thank you for your correction.

In the title, chemical formula “PbS” should be added

According to this suggestion, PbS was replaced instead of lead sulfide.

Reference [1] is over 20 years old, as it is the first reference that should introduce the reader with the topic. This reference should be replaced by a recent review paper (or a few).

Thank you for your comments: we agree. We added a review for microbial synthesis of semiconductor metal sulfide nanoparticles (M. Raouf Hosseini & M. Nasiri Sarvi,  Materials Science in Semiconductor Processing 40 (2015) 293–301) in L45-46. However, we referred Reference[1] as a reference of biological induced biomineralization. Therefore, we would like to remain it.

References 3 and 4 are very old, i.e. ref. 4 is from 1989! A lot has changed since that time, and this should be updated.

According to this suggestion, we added a recent publication for PbS (Wei et al. Scientific Reports (2021) 11:1216) in L52-55. References[3,4] described pricise observation that correspond to our results, meaning these are old but respectful studies as original papers. Thus, we would like to remain them, too.

Line 52, CdS would precipitate from the solution containing the Cd(II) and S(-II) ions anyway, so what is the benefit of adding the bacteria to that solution? It must be stated clearly in the introduction.

Thank you for your suggestion. This work showed Cd2+ and S2- made spontaneously precipitates according to ksp, however INTRACELLULAR CdS nanoparticles showed precisely controlled in size. They did not mention about extracellular CdS, but Cd2+ and S2- were incorporated into the cells and intracellular organic substances like glutathione were hypothesized for 2-5 nm sized particle formation. The explanation “Moreover, the authors hypothesized intracellular thiol or glutathione contents were related to nanosized control (2-5 nm) . This report indicated that spontaneously formed CdS could be maintained as a nanosized particle due to the bacterial organic matters. ” was added in L.60-63.

Line 77, it should be 10.50±0.15

Thank you for your correction.

The Ksp of lead sulfide is 8.0 × 10−28, so what’s the reason of adding the bacteria as this salt will precipitate anyway? Is it about the size of the crystals formed? This must be stated clearly in the introduction.

Yes. As mentioned above, bacterial proteins were hypothesized as the size control factor. The sentence in L.60-63 can explain the reason why using bacteria.

Line 106, it should not be Pb but Pb(II) or Pb2+

Thank you for your correction.

Line 151, what do you mean by “sufficient quality”?

The materials of sufficient quality mean quality which can show the general fundamental characteristics of a semiconductor.

Figure 3, C and F, please explain why the signals from Cu are more intensive than those from Pb? Where this Cu comes from?

Cu comes from 150-mesh copper grids coated with collodion (Materials & Methods L.381-382).

Fig. 3C and 3F showed cell images, thus, background Cu intensity is dependent on cellular thickness. Pb intensity from intracellular PbS was also affected by cellular thickness.  

Line 186, “Crystal lattice” can’t be observed, this is too colloquial. Please rephrase.

We are sorry for using the phrase. We rephrased that “lattice images”.

Why the pH was not monitored in those studies? This is crucial as it affects the formation of PbS.

RCVBN medium contains buffer component (phosphate buffer) and basically maintained at pH 7.6. Culture supernatant after bacterial growth showed only slight change.

Line 226, it should be “low purity”

We are sorry for the confusion we may cause.

We meant “bacterial PbS showed relatively pure compared with chemically synthesized PbS” indicated in L223-227.

Figure 5, why the Authors compare the regions between 2-theta 54 and 66? This region is not informative at all, due to the very low intensities (compared to the noise level). Instead, the Authors should compare the other range, 5-20 deg. Please provide “separate” grey and red patterns as I can see a lot of peaks at the grey (synthetic) one that do not appear at the red one. This must be explained as one may doubt that the correct phase of PbS has been obtained.

We determined thin film XRD with 2θ between 20° and 70° to confirm the impurities. For the region between 54 and 66, magnified drawing was inserted according to the suggestion by another reviewer’s comment. However, we agree with your comment that this region is not informative. Therefore, we deleted the insert and newly made comparative patterns of 5 – 20 degrees, instead. Thank you very much for fruitful comment.

We could not identify the impurities in chemically synthesized PbS.

Line 449, please provide accession number

We have deposited the sequence appropriately. We are waiting for releasing accession numbers from DDBJ after detail checking.

We would like to add some corrections we realized. (indicated by yellow highlight)

Equation (1) did not include x which is light travel distance in a medium (thickness of sample). In fact, x of PbS region cannot be determined precisely in our samples, therefore, actual calculated values of α2 were not shown on Y-axis in Fig. 6. However, graph shape of α2 is independent of x and the tangent is same as previous figure.

Moreover, we did not show the data of extracted PbS from the cells grown with 5mM lead acetate in L290-293. However, we would like to show the graph in supplemental data in this revised manuscript for the readers. 

Round 2

Reviewer 2 Report (New Reviewer)

The Authors have revised and improved their manuscript. This version can be accepted.

This manuscript is a resubmission of an earlier submission. The following is a list of the peer review reports and author responses from that submission.

Round 1

Reviewer 1 Report

The manuscript authored by Okamura et al reports about the characterization of biogenic lead sulfide quantum dots derived from bacterial consortia. The development of green nanomaterial synthesis course is currently highly-demanded, the results obtained herein would be fascinating the reader in this journal, IJMS.

I feel this research concept has a potential to open the door for new quantum dots synthesis process under ambient condition, although further basis researches including the synthesis condition optimization, elucidation of metabolism, and property tuning etc should be addressed in the future. Overall, the manuscript is very concise, well written and easy to follow. The experimental strategy is clearly explained, all relevant details are provided to understand the methodology and the value of results.

I have no critical issues with the manuscript but I do have an editorial suggestion to add conclusion section with your future perspectives as I mentioned above.

I would suggest you to check all English by native speaker.

Author Response

Answers to reviewers’ comments

Reviewer #1

Comments and Suggestions for Authors

The manuscript authored by Okamura et al reports about the characterization of biogenic lead sulfide quantum dots derived from bacterial consortia. The development of green nanomaterial synthesis course is currently highly-demanded, the results obtained herein would be fascinating the reader in this journal, IJMS.

I feel this research concept has a potential to open the door for new quantum dots synthesis process under ambient condition, although further basis researches including the synthesis condition optimization, elucidation of metabolism, and property tuning etc should be addressed in the future. Overall, the manuscript is very concise, well written and easy to follow. The experimental strategy is clearly explained, all relevant details are provided to understand the methodology and the value of results.

I have no critical issues with the manuscript but I do have an editorial suggestion to add conclusion section with your future perspectives as I mentioned above.

We appreciate all of the helpful suggestions by the reviewer.

We added Conclusion section to emphasize our conclusion.

Comments on the Quality of English Language

I would suggest you to check all English by native speaker.

Both original and this revised manuscript have been proofread by BioScience Writers, LLC, Texas, USA.

Reviewer 2 Report

Is extracellular crystal growth unrelated to intracellular crystals? If the authors believe (I would like to know more about localization) that sulfate reduction (S2−) occurred on the cell membrane, and subsequently, PbS formed through an ionic reaction with Pb2+, does this mean that crystals of two sizes are formed in the cells? 22 nm and small? It is possible that the SO42- into S2- reaction process is biogenic, but the PS formation reaction can be extracellular with subsequent aggregation and formation of crystalline structures. This can also explain the dependence of microcrystal sizes on the amount of lead acetate added.

Author Response

Answers to reviewers’ comments

Reviewer #2

Comments and Suggestions for Authors

 Thank you very much for reviewing our manuscript and providing valuable discussion.

Is extracellular crystal growth unrelated to intracellular crystals? If the authors believe (I would like to know more about localization) that sulfate reduction (S2−) occurred on the cell membrane, and subsequently, PbS formed through an ionic reaction with Pb2+, does this mean that crystals of two sizes are formed in the cells? 22 nm and small?

Based on this comment, we added a detailed explanation of the intracellular particles in the Results section (Line 156-160). Peaks of C, O, and P were present in the EDS analysis of the intracellular particles, suggesting that the presence of phosphate caused the intracellular PbS aggregates to become bigger than the external particles. The intracellular aggregates were not clearly observed by TEM because of the cell membrane, so we did not find that the intracellular aggregates consisted of 5 nm particles or not.     

It is possible that the SO42- into S2- reaction process is biogenic, but the PS formation reaction can be extracellular with subsequent aggregation and formation of crystalline structures. This can also explain the dependence of microcrystal sizes on the amount of lead acetate added.

Thank you for this comment. We agree that the PbS formation can be extracellular and that this can explain the dependence of the crystal size on the amount of lead acetate added.

Reviewer 3 Report

The manuscript by Y. Okamura et al. describes the formation and properties of PbS nanoparticles formed by bacterial biomineralization. The nanostructures were investigated by XRD, and optical and electrical measurement techniques, revealing semiconducting behavior resembling PbS quantum dots. The study is interesting. However, there are several unclear points, which I think the authors must enhance discussions in these directions. 

First of all, the introduction does not give proper information about bacterial synthesis. I suggest the authors add more notes in the introduction to explain this process, considering more important/demanding parameters.

I also suggest making a separate section for synthesis and sample preparation.

One of the main concerns regarding applying colloidal QDs is their stability over time and against specific environments. I suggest the authors extend discussions in this regard. The following review paper can be interesting. https://doi.org/10.1016/j.mtsust.2022.100305

The text (information) in Fig 2 is not redeable. The figure must be improved.

Also, the peaks in the XRD pattern should be indexed.

I suggest the authors read the text carefully and eliminate repetitive statements/discussions and typos/errors.

The conclusion is missing. The results must be shortly re-emphasized in the conclusion. 

The manuscript contains several grammatical errors that must be addressed before the final decision. Also, the flow of the text should be optimized.   

Round 2

Reviewer 2 Report

The authors changed the text by adding references to publications on this topic in the Introduction, a description of the production of lead sulfide in the Methods section, a few more text inserts, and a Conclusion section. These fragments are highlighted in red, however, when compared with the original text, it is clear that there are still some additions and cropping. Despite these changes, the text has not fundamentally changed. In the first version of the review of the manuscript, I asked 8 questions, I send them again with numbering. Unfortunately, the authors did not answer them, except for a partial response to question No. 2, namely, they improved the quality of the drawing, now it can be read.

The authors did not explain why such samples of microbial communities were taken (question No. 1).

The composition of the RCVBN nutrient medium is not indicated, which is very important in this work (question no. 3).

When describing anaerobicity and facultative anaerobicity (lines 119-129), the method of cultivation on agar layers is not given (question No. 4).

The authors suggest that it is possible to develop a technology for the production of lead sulfide for the manufacture of semiconductors, the greater the requirements for the stability of a consortium of microorganisms. The conditions for maintaining viability and composition, maintaining percentages (Fig. 2) are not described, and the conclusion about the stability of these two microbial consortia is unjustified. (No answer to question #6).

Questions 5, 7 and 8 are the most fundamental, and the authors did not answer them either. The formation of lead sulfide nanoparticles may be the result of an extracellular chemical process. I will add that micrographs obtained by the described TEM method are not proof of intracellular crystal formation. They are more reminiscent of the sorption of crystals on the surface of cells.

Unfortunately, the article cannot be published in this form.

Questions, in the previous review of the first draft of the manuscript:

In the submitted manuscript, the authors investigated the possibility of obtaining lead sulfide nanoparticles using bacteria: “In this study, we screened bacteria that form PbS nanoparticles and characterized the crystallinity and semiconductor properties of the resulting particles.” The task of the proposed implementation was not so much about wastewater treatment, but about obtaining quantum dots of lead sylphide and the possibility of their practical application. Accordingly, the experiment was carried out by specialists of two profiles - biologists and physicists.

1. The very concept of "screening" means the analysis of a large number of objects and the choice of the necessary. In this case, the work was carried out only on two samples isolated from the seabed, containing sea sponges and gravel. The authors did not explain why they took these particular samples from nature. Maybe it would be more expedient to take soil samples, for example, in places where lead was mined?

2. It was not possible to isolate individual bacteria, so the work was carried out with two mixed samples of PS-1 and PS-2 (consortiums), and the identification of microorganisms was carried out to a family or genus based on reading fragments of the 16S rRNA gene. The authors do not specify at what stage the taxonomic assessment of the composition of microorganisms was made, probably after obtaining enrichment cultures. Figure 2 shows the percentages of families and genera of microorganisms in consortiums. Unfortunately, this drawing was made carelessly. Even at high magnification, not all names can be read, or rather guessed. The text provides a list of taxa: "... Pseudomonas (10.4%), Marichromatium (7.8%), S. algae (6.1%), Desulfovibrio (5.7%),...". Probably "S. algae" means a group of seaweeds, it is not a taxon, but a mixed group of microorganisms.

3.  Enrichment cultures were obtained in order to increase the content of PbS-forming bacteria, for which purpose the nutrient medium "RCVBN medium [25]" was added to the samples. The composition of the RCVBN medium is not specified, although this is important in this work. Below in the text is a list of metals included in this medium as trace elements. To understand the full medium composition, it is proposed to look at the link [25]. Unfortunately, in this work there is no recipe for the RCVBN medium, but there are two following references [13 and 14], articles on which are not in the public domain. I managed to see one of them, there is also no recipe for the medium.

4. In the Results section it says: "Therefore, the bacteria could form colonies on the agar covering the soft agar (the agar overlay or top agar) but not grow in nitrogen-purged liquid medium on a conventional clean bench." I would like an explanation of this experiment, besides, there are no methods of cultivation on agar medium. 

5. “The PbS-forming bacteria showed a black color derived from minerals; therefore, it can be distinguished easily from other bacteria. Without lead acetate, the enrichment cultures PS-1 and PS-2 were red, indicating photosynthetic bacteria. When supplemented with lead acetate, the culture color darkened with increasing lead acetate concentration (Fig. 1)." Since the work was carried out with consortiums, the color change in test tubes could be explained by the death of some microorganisms and the safety of others. In both samples, at a concentration of 5 mM lead cations, the concentration of the red pigment was increased, and at 10 mM, the red color almost disappeared. A succession of the species composition is possible, which would be worth studying and trying to identify active species according to the criterion for the formation of biogenic lead sulfide. There is no convincing information that part of the lead sulfide is formed biogenically, and not due to a chemical reaction.

6. «Bacterial consortia that stably formed PbS nanoparticles were obtained.» What is the stability over time? How many passages did PS-1 and PS-2 survive? How has the proportion between different groups of microorganisms changed?

7. The manuscript states that: «Extracellular particle size ranged from 3.9 to 5.5 nm. … However, the morphology and size differed between extracellular and intracellular particles. The intracellular PbS particles were spherical, distributed over the cell, and had a uniform size of around 22 nm. …….. extracellular particles occurred at a specific site on the cell membrane, and extremely small granules were pushed outward and aggregated. Moreover, the extracted particles observed at high magnification showed that the aggregates consisted of particles a few nanometers in diameter. Fig. 4 shows that 10 nm microcrystals grew with 1mM lead acetate, but 3.9–5.5 nm QD-sized particles grew with 500 μM lead acetate…..». If the granules in the cells are round and 22 nm in size, then which "extremely small granules were pushed outward and aggregated"? In Figure 4C, the cubic structure of the crystals is guessed, similar to galena.

8. Is extracellular crystal growth unrelated to intracellular crystals? If the authors believe (I would like to know more about localization) that sulfate reduction (S2−) occurred on the cell membrane, and subsequently, PbS formed through an ionic reaction with Pb2+, does this mean that crystals of two sizes are formed in the cells? 22 nm and small? It is possible that the SO42- into S2- reaction process is biogenic, but the PS formation reaction can be extracellular with subsequent aggregation and formation of crystalline structures. This can also explain the dependence of microcrystal sizes on the amount of lead acetate added. («Fig. 4 shows that 10 nm microcrystals grew with 1mM lead acetate, but 3.9–5.5 nm QD-sized particles grew with 500μM lead acetate.»).

There are a lot of assumptions in this manuscript, although the idea of obtaining a semiconductor material suitable for industrial use, while purifying the medium from heavy metals (lead in this case) seems to be very interesting.

Reviewer 3 Report

I find the authors' response inadequate and insufficient. The authors have surrounded the questions instead of direct explanations. Also, there are many inconsistencies between the results and explanations. The work itself is very similar to previously published works and does not add any news to the field. As stated in my previous review, the manuscript contains some errors and hard-to-understand sentences. Moreover, the revised version has even more bac-and-force statements, lengthy sentences, and very poor scientific statements alongside extremely basic discussions. I don't recommend its publication in the International Journal of Molecular Sciences, which is a highly ranked journal and should avoid such less interesting works. 

As stated in my previous review, the manuscript contains some errors and hard-to-understand sentences. Moreover, the revised version also suffers from bac-and-force statements, lengthy sentences, and very poor flow.